# Retrospective Clinical Analysis of Epilepsy Treatment for Children with Drug-Resistant Epilepsy (A Single-Center Experience)

**DOI:** 10.3390/brainsci13010014

**Published:** 2022-12-21

**Authors:** Changqing Liu, Yue Hu, Jian Zhou, Yuguang Guan, Mengyang Wang, Xueling Qi, Xiongfei Wang, Huawei Zhang, Aihemaitiniyazi Adilijiang, Tiemin Li, Guoming Luan

**Affiliations:** 1Department of Neurosurgery, Sanbo Brain Hospital, Capital Medical University, Beijing 100093, China; 2Beijing Key Laboratory of Epilepsy, Sanbo Brain Hospital, Capital Medical University, Beijing 100093, China; 3Center of Epilepsy, Beijing Institute of Brain Disorders, Capital Medical University, Beijing 100093, China; 4Department of Neurosurgery, Aviation General Hospital, China Medical University, Beijing 100012, China; 5Department of Neurology, Sanbo Brain Hospital, Capital Medical University, Beijing 100093, China; 6Department of Pathology, Sanbo Brain Hospital, Capital Medical University, Beijing 100093, China

**Keywords:** drug-resistant epilepsy, children, resection surgery, palliative surgery, seizure outcome

## Abstract

Objectives: This retrospective cohort study investigated the clinical characteristics and seizure outcomes of patients aged 1–14 years with drug-resistant epilepsy (DRE) who were treated by different typologies of therapy. Methods: Four hundred and eighteen children with DRE were recruited from Sanbo Brain Hospital of Capital Medical University from April 2008 to February 2015. The patients were divided into three groups: medication (*n* = 134, 32.06%), resection surgery (*n* = 185, 44.26%), and palliative surgery (*n* = 99, 23.68%) groups. Demographic characteristics were attained from medical records. All patients were followed up for at least 5 years, with seizure outcomes classified according to International League Against Epilepsy criteria. The psychological outcome was evaluated with the development quotient and Wechsler Intelligence Quotient Scale for children (Chinese version). Results: The most frequent seizure type was generalized tonic seizure in 53.83% of patients. Age at seizure onset in 54.55% of patients was <3 years. The most frequent etiologies were focal cortical dysplasia (FCD). West syndrome was the most common epilepsy syndrome. Favorable seizure outcomes at the 5-year follow-up in the medication, resection surgery, and palliative surgery groups were 5.22%, 77.30%, and 14.14%, respectively. The patients showed varying degrees of improvement in terms of developmental and intellectual outcomes post-treatment. Conclusions: Pediatric patients with DRE were characterized by frequent seizures, a variety of seizure types, and complex etiology. Recurrent seizures severely affected the cognitive function and development of children. Early surgical intervention would be beneficial for seizure control and prevention of mental retardation. Palliative surgery was also a reasonable option for patients who were not suitable candidates for resection surgery.

## 1. Introduction

Epilepsy is among the most common chronic neurological disorders affecting the quality of life in patients. Children constitute the majority of epilepsy patients with an annual incidence rate of 41–187/100,000, which is much higher than that of adults [1]. In China, this ratio reaches 151/100,000 [2]. The causes of seizures are diverse and are generally classified as genetic, structural and metabolic, and unknown [3].

Antiepileptic drugs (AEDs) are the main form of treatment. About 70% of patients respond well to AEDs and achieve seizure-free outcome. In 20%–30% of cases, however, seizures remain uncontrollable [4,5,6]. The International League Against Epilepsy (ILAE) defined drug-resistant epilepsy (DRE) as a failure of adequate trials of two tolerated and appropriately chosen and used AED schedules (whether as monotherapies or in combination) to achieve sustained seizure freedom [7]. In such cases, surgical intervention is widely used as the primary treatment for pediatric epilepsy [8]. A large histopathological study showed that the main pathology of DRE in children is cortical malformations and tumors. This is clearly different from the pathology in adults, which is hippocampal sclerosis [9]. Notably, not all patients are suitable for resection surgery. For patients with epileptogenic regions involving eloquent cortical areas or diffuse lesions, multiple subpial transections (MST) and corpus callosotomy were the main palliative procedures to reduce seizures in the last century [10]. Due to advances in knowledge and medical technology, neuromodulation for epilepsy is emerging as treatment for epilepsy [11,12]. Regardless of the surgical approach, the goal is to reduce the propagation of epileptiform discharges.

Epilepsy in children with cognitive decompensation was formerly referred to as catastrophic epilepsy although it was not a clinical classification [13]. Furthermore, the term was not recognized by the International League Against Epilepsy (ILAE) [3]. Gradual and severe seizures in children affect their daily life, especially academic performance [14,15,16]. Thus, DRE is a major public health problem given the significant negative impacts on children’s development [17]. Effective treatment of DRE while minimizing the side effects remains a clinical challenge.

However, few studies have systematically compared the efficacy of different typologies of therapy in children with DRE. The aim of our study was to generalize the clinical characteristics of children with DRE. Our hypothesis was that surgical resection would bring children with DRE the most favorable seizure outcomes, and followed by palliative surgery. Good seizure outcomes were associated with improved cognitive function. Moreover, analysis of predictive factors in patients undergoing resection surgery would guide clinical practice. Additionally, this study evaluated long-term cognitive outcomes of children with DRE. To the best of our knowledge, this series reported the largest single-center cohort study on different treatments for pediatric epilepsy.

## 2. Materials and Methods

### 2.1. Patient Selection

We retrospectively analyzed the data of 516 patients with DRE who were admitted to Sanbo Brain Hospital of Capital Medical University from April 2008 to February 2015. The inclusion criteria for patients were as follows: (1) aged 1–14 years; (2) uncontrolled seizures after AED and adrenocorticotropic hormone therapy; (3) had undergone examinations by magnetic resonance imaging (MRI), computed tomography (CT), and video electroencephalogram (VEEG); (4) follow-up data were available; (5) informed consent by guardians. The exclusion criteria: (1) Benign epilepsy syndromes in children such as benign occipital epilepsy, benign childhood epilepsy with centro-temporal spikes, and other syndromes which respond better to AEDs; (2) if it was not clear whether the seizures were recurrent or the epileptogenic foci had not been completely removed (Figure 1); (3) patients with a history of surgical resection. The study was approved by the ethics committee of Sanbo Brain Hospital, Capital Medical University.

### 2.2. Psychological Assessments

The choice of psychological test used was subject to the age and cognitive level of the patients. A Chinese version of the Gesell Developmental Schedules (GDS) or Wechsler Intelligence Quotient Scale (WIQS) was used for psychological assessments of patients who met the inclusion criteria [18,19]. The Chinese version of GDS is a widely used classic psychometrical scale for evaluating neurodevelopmental outcomes of children less than 6 years old. The scale assesses five areas: gross motor skills, fine motor skills, adaptability, language, and social activity. Each area yields a developmental quotient (DQ) [20]. The total DQ is the average of the five DQs mentioned above. DQ was categorized as normal (≥80), borderline delayed (≥70, <80), mildly delayed (≥50, <70), moderately delayed (≥35, <50), severely delayed (≥20, <35), and extremely delayed (<20). WIQS scores were used to evaluate the children’s intellectual quotient. The WISC revised in China (C-WISC) was used for children aged 6–16 years old. The C-WISC covers full scale IQ, verbal IQ, and performance IQ. Our analysis classified them as normal (≥90), mild defect (≥70, <90), moderate defect (≥60, <70), and extreme defect (<60).

### 2.3. Demographic and Clinical Characteristics

Preoperative evaluations were seizure semiology; a detailed history; neurologic examination; long-term VEEG, MRI, and CT. The seizure type was classified according to the International Classification of Epileptic Seizures; the syndrome type was defined according to the International Classification of Epileptic Syndromes; etiology was determined based on brain MRI and CT and cytogenic findings as well as psychological development at the time of the first admission. If the location of epileptic foci was not clear from the above results, magnetoencephalography (MEG) and positron emission tomography (PET) data were examined. The cases were discussed by a multidisciplinary team of epilepsy neurosurgeons, neuropsychologists, and neuroradiologists at our epilepsy center. The patients’ condition and treatment options were explained to the family, and the treatment was decided by the children’s guardian(s). All patients were followed up at the outpatient clinic or by telephone interview for at least 5 years. Seizure outcomes were evaluated based on the ILAE classification [21]: ILAE class 1, completely seizure free with no auras; ILAE class 2, only auras and no other seizures; ILAE class 3, one to three seizure days per year with or without auras; ILAE class 4, four seizure days per year to 50% reduction of baseline seizure days with or without auras; ILAE class 5, less than 50% reduction of baseline seizure days to 100% increase of baseline seizure days with or without auras; ILAE class 6, more than 100% increase of baseline seizure days with or without auras. ILAE classes 1 and 2 were defined as favorable and ILAE classes 3–6 as unfavorable outcomes. Resection surgeries included epileptic focal resection, lobotomy, multilobar resection, and hemispherectomy. Palliative surgeries included corpus callosotomy, MST, and vagus nerve stimulator implantation. Patients in whom noninvasive tests could not accurately localize the epileptogenic focus, such as MRI showing multifocal brain abnormalities or negative, some of the patients would undergo stereotactic EEG. If the patient was not suitable for resection surgery after a comprehensive evaluation, we would use intracranial electrodes to perform radiofrequency thermocoagulation (RF-TC). Therefore, RF-TC was considered another palliative procedure.

### 2.4. Statistical Analysis

Psychological assessments at the 1st, 3rd, and 5th year follow-ups were performed to evaluate the cognitive or behavioral problems. In children with non-resective surgery, the palliative or medication groups, clinical characteristics, and prognosis were compared. Continuous variables were described as means and standard deviations, and categorical variables as frequencies and percentages. To identify potential predictors of seizure outcomes in patients treated by surgery, a univariate analysis was carried out, where categorical variables were analyzed with the chi-squared or Fisher’s exact test. Continuous variables were compared with the Mann–Whitney U test. Variables with a *p* < 0.2 in the univariate analysis were entered into a multivariate logistic regression model by backward elimination. Statistical significance was considered at *p* < 0.05. Analysis was performed using SPSS version 25.0 software (SPSS Inc., Chicago, IL, USA).

## 3. Results

### 3.1. Demographic Characteristics

A total of 516 patients met the inclusion criteria. During follow-up, 55 patients were excluded for changing their treatment regimen at enrollment and a further 43 were lost to follow-up. Hence, 418 patients, 272 males, and 146 females were included in the analysis (Figure 1). The age at seizure onset was <6 months in 82 patients (19.62%), 6 months–1 year in 45 (10.77%), 1–2 years in 52 (12.44%), 2–3 years in 49 (11.72%), and >3 years in 190 (45.45%) patients (Figure 2).

### 3.2. Seizure Types

A total of 160/418 (38.28%) patients had one type of seizure. Generalized tonic seizure (GTS) was the most common seizure type at 53.83%, complex partial motor at 43.06%, generalized tonic-clonic seizures at 30.62%, and simple partial motor at 20.57%. Partial seizures were recorded in 236 patients, and 68 of them had a secondary generalized tonic-clonic seizure.

### 3.3. Etiology

The etiology of epilepsy was determined based on the medical records, CT, brain MRI, and cytogenic findings. The five common etiologies were focal cortical dysplasia (*n* = 79, 18.90%); hypoxic-ischemic encephalopathy (*n* = 64, 15.31%); tumors (*n* = 33, 7.90%); central nervous system infection (*n* = 29, 6.94%); and Rasmussen encephalitis (RE; *n* = 22, 5.26%). The etiology in 101 patients (24.16%) was unknown.

### 3.4. Epileptic Syndrome

Unclassified epilepsy was the frequently observed type of epilepsy (*n* = 196, 46.89%), followed by West syndrome (WS; *n* = 78, 18.66%), Lennox-Gastaut syndrome (LGS; *n* = 67, 16.03%), neurocutaneous syndromes (NS; *n* = 28, 6.70%), and RE (*n* = 22, 5.26%) (Figure 2).

### 3.5. Psychological Assessment

DQ was evaluated in 135 patients. Normal patients were 25/135 (18.52%), borderline delayed 9/135 (6.67%), mildly delayed 22/135 (16.30%), moderately delayed 26/135 (19.26%), severely delayed 34/135 (25.19%), and extremely delayed 19/135 (14.07%).

The remaining 283 patients were evaluated with the C-WIQS; normal 30/283 (10.60%), mild retardation 97.283 (34.28%), moderate retardation 51/283 (18.02%), and severe retardation 105/283 (37.10%). Thus, cognitive ability was normal or borderline delayed in 25.19% and 10.60% of patients, respectively (Figure 3).

### 3.6. Seizure and Psychological Outcomes

Seizure outcomes were evaluated based on the ILAE classification. The three treatment groups were 134 patients (32.06%) on medication, 185 (44.26%) with resection surgery, and 99 (23.68%) with palliative surgery. Although the prognosis of medication should not be assessed with this type of classification, the seizure outcomes were also evaluated with the ILAE classification of patients in the medication group for comparison with the outcome of surgical treatment. Favorable seizure outcomes were observed in 14.93%, 81.62%, and 19.19% of patients on medication, resection surgery, and palliative groups, after 1-year follow-up; 10.45%, 78.92%, and 16.16% after 3-year follow-up; 5.22%, 77.30%, and 14.14% after 5-year follow-up. Cognitive function was evaluated based on DQ and C-WIQS scores. The use of DQ was limited by age, hence, only 63 patients less than 3 years old were followed up at the time of the first admission for three years. Favorable psychological outcomes were observed in 61.11%, 70.97%, and 71.43% of patients in the medication, resection surgery, and palliative surgery groups, respectively, in the children younger than 6 years old. Patients older than 6 years showed favorable psychological outcomes in 71.13%, 81.88%, and 82.50%, respectively, at the last follow-up. The mean number of antiepileptic drugs taken by patients in the medication, resection surgery, and palliative surgery groups at the time of enrollment in this study was 2.32, 2.18, and 2.58, respectively. At the last follow-up, this figure was 2.13, 1.45, and 1.85, respectively. All patients in the medication group were still taking AEDs, while 43 and 7 in the resection and palliative surgery groups, respectively, were off AEDs (Figure 4 and Figure 5).

## 4. Discussion

The lifetime prevalence of epilepsy has been estimated to be 7.60 per 1000 persons [22], and the global prevalence of active epilepsy is in the range of 4.9–12.7 per 1000 [23]. Children constitute the majority of epilepsy patients, with an annual incidence of 41–187 per 100,000, which is higher than that of in adults [1]. Most intractable epilepsy syndromes such as WS, LGS, NS, and RE are the main causes of DRE [24]. The onset of seizures typically occurs at a young age. In the present study, the earliest onset occurred at 3 days after birth. DRE is often accompanied by developmental delay, systemic comorbidities and has a high risk of mortality. The clinical characteristics of children below 6 years old who developed DRE were investigated [25]. However, the association between different types of therapy and seizure outcomes has not been reported. Few large cohort studies have compared long-term seizure outcomes and cognitive function outcomes following different treatment methods in DRE patients. Based on cognitive function and seizure control in epileptic pediatric patients, surgical treatment should be carried out early in DRE patients, but surgery was not fully utilized [26]. Here, we outline the clinical features of childhood refractory epilepsy, and compare the effects of different treatment modalities on the prognosis and cognitive function of the children.

Drug-resistant epilepsy, as the name and definition suggest, has poor effect on AEDs. Previous studies showed that the probability of achieving seizure-free by tweaking drug regimens after failure of two AEDs was less than 10% [27]. Figure 5 shows treatment with newer AEDs yielded satisfactory seizure outcomes in only a minority of patients. Given that the lobes of the brain are interconnected via fibers, repeated epileptic seizures can cause secondary pathologic changes [28]. For instance, lesions located in the temporal lobe may lead to secondary hippocampal sclerosis [29]. Pediatric epilepsy surgery aims to remove the epileptogenic area or limit the propagation of epileptiform discharges in the brain. Clinically, the application of surgical treatment for DRE in children is increasingly reported. In our epilepsy center, resection surgery was recommended for suitable patients. In a randomized controlled trial, Dwivedi et al. reported that with the rate of seizure freedom was 7% and 77% in the medication group and surgical treatment group respectively [30].

Epileptic syndromes begin in childhood that have different ages at seizure onset [31]. However, not all children with epilepsy can be classified to have a specific epilepsy syndrome. In the present study, unclassified epileptic syndromes were observed in 46.89% of cases (Figure 2). Excluding those that could not be classified, WS was the most common epilepsy syndrome in this study, accounting for 18.67% of all patients. It was followed by LGS (some patients had a history of WS), which accounted for 16.03%. The peak age at seizure onset in WS has been reported to be 5 months [29]. In our cohort, 67.95% of patients with WS had seizure onset in the first year of life, and 20 were born with hypoxic-ischemic encephalopathy. Children with WS usually have unfavorable seizure outcomes [32,33] and most have mental retardation that evolves into LGS. In general, only 25.64% of patients with WS attained favorable seizure outcomes. LGS is another common childhood epilepsy syndrome, which is characterized by a triad of multiple seizure types, intellectual impairment, and a specific EEG pattern of slow-spike waves, with or without paroxysmal fast activity in sleep. However, not all patients had the three features at the onset of the seizure [34]. Over time, cognitive impairment would become more pronounced. Additionally, uncontrolled seizures would contribute to further worsening of cognitive impairment. Therefore, it had been suggested that the diagnostic criteria should not be so strict to allow patients to receive novel treatment early [35]. Pediatric patients might have multiple seizure types. In the present study, the most common seizure types were generalized tonic seizures. The capture of multiple seizure types was helpful in the diagnosis of epilepsy syndrome.

The major difference between children and adults was that children’s brains were not yet fully developed. In this study, the age of seizure onset was seen in more than 50% of children younger than three years old, which is a critical period of brain development. Recurrent seizures would cause the occurrence of ischemia and hypoxia in neuronal tissue. It might affect the normal development of brain tissue. Darra et al. reported most severe cognitive dysfunction and motor impairment in patients were associated with persisting seizures [36]. Effective control of seizures might benefit cognitive function and motor function. Moreover, children with DRE with severe functional impairment from the onset of early age seizures or long-duration seizures show abnormal neurologic and psychological development, even as they become seizure-free [37].

In this cohort study, we compared the demographic and clinical characteristics of patients in the medication and palliative surgery groups. The results showed no statistically significant differences between the two groups for these basic characteristics (Table 1). However, the seizure outcome of patients in the palliative surgery group was significantly better than the medication group. It had been reported that difficult localization of epileptogenic zones, such as bilateral epileptic discharges or no positive MRI findings, may be related to unfavorable seizure outcomes [38]. Moreover, the lesions that overlap with the functional cortex may be linked with the unfavorable seizure outcomes. Patients who show the above preoperative examination results might likely not undergo resection surgery. Palliative surgical procedures such as corpus callosotomy, bipolar electrocoagulation, and neuromodulation are effective for patients without indications for resection surgery [11,12,39,40,41]. Baba et al. reported a group of 56 patients with refractory WS who underwent corpus callosotomy and were seizure-free after surgery in 32.1% of patients [42]. A previous review showed that vagus nerve stimulation resulted in 6%–27% of patients with refractory epilepsy being seizure-free [43].

A total of 185 patients with DRE underwent resection surgery. The rate of seizure freedom at the last follow-up was 77.30% (Table 2), which was similar to studies by studies [25,44,45]. Longer duration of seizures was significantly associated with unfavorable seizure outcomes [46,47]. Similarly, the present study also found longer duration of seizures was an independent predictor on unfavorable seizure outcome. In this group, there was no significant association between seizure outcomes and age at surgery (Table 3). In this study, multivariate analysis revealed that unilateral ictal onset rhythm was a predictor of seizure-free. Although 23 patients in this group had infrequent seizures that were not captured, 22 patients had definite lesions on the MRI. Pathologically, FCD, tumors, and hypoxic-ischemic encephalopathy were the most common etiologies and were observed in 38.38%, 17.84%, and 10.27% of cases, respectively. In a European study of 9523 patients who underwent epilepsy surgery, the most common cause of childhood epilepsy was FCD, followed by tumors [9]. In another study of 543 children who underwent epilepsy surgery, 60% were <2 years old and the most frequent etiologies were cortical dysplasia (42.4%), tumors (19.1%), and atrophy/stroke (9.9%) [48]. FCD and tumors are regional and structural abnormalities that are difficult to visualize by MRI at an early stage [49,50]. As a result, patients with such lesions often show a poor response to AEDs, and are likely to be classified as DRE [51,52].

During the follow-up, the intellectual and psychological development of most patients showed varying degrees of improvement (Figure 4). The most favorable outcomes for both seizures and cognitive function were in the resection group, the palliative surgery, and the medication groups respectively (Figure 5), similar to previous reports [53,54]. A study conducted by Veersema et al. [55] showed that 2 years post epilepsy surgery 24 of 36 patients had a ≥10 points increase in IQ. These were similar to our study, where patients who had resective epilepsy surgery had an IQ increase of ≥10 points (25.95%) in the resection group. Similar results were recorded in 16.16% and 6.72% of the palliative surgery group and medication group, respectively. Cumulatively, the side effects of long-term AED use diminish efficacy and lead to mental retardation [56,57]. In the present study, 43 and 7 patients in the resection and palliative surgery groups, respectively, were successfully weaned off AEDs. In a systematic review of epilepsy in children, a strong relationship was found between intelligence quotient decline and continued seizures [58]. Moreover, a cohort study of children <3 years old who underwent epilepsy surgery found that early surgery was associated with increased postoperative DQ [59]. As a consequence, evidence shows that improvement in cognitive function was closely linked to better seizure outcomes.

The limitations of our study were: first, it was a single-center retrospective analysis in which not all variables could be controlled, for example, patient selection and time of surgery. Second, seizure outcomes were determined from the medical records of patients or by telephone interview without clinical assessment, which may have introduced recall bias. Future studies should consider factors such as genetic data, and family history of epilepsy. Third, detailed information was obtained from the medical records. Therefore, we did not adopt the 2017 classification of seizure types since the earliest cases enrolled in this study was from 2008.

## 5. Conclusions

The results of this study demonstrated that compared with adult patients, children with DRE were a special group with their own characteristics. Based on seizure outcomes and cognitive function, surgical resection seemed to be the most effective treatment. Short duration of seizure and unilateral ictal rhythm were associated with favorable seizure outcomes. Palliative surgery could still benefit patients for who resection surgery was not an option. Regardless of the surgical treatment regimen, children with DRE should be treated as early as possible to prevent progressive brain damage and mental retardation caused by epilepsy.

## Figures and Tables

**Figure 1 brainsci-13-00014-f001:**
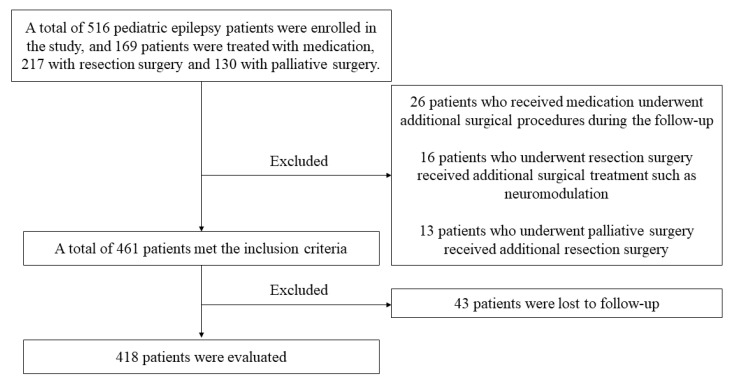
Flowchart describing the procedures and exclusion and inclusion criteria of this study.

**Figure 2 brainsci-13-00014-f002:**
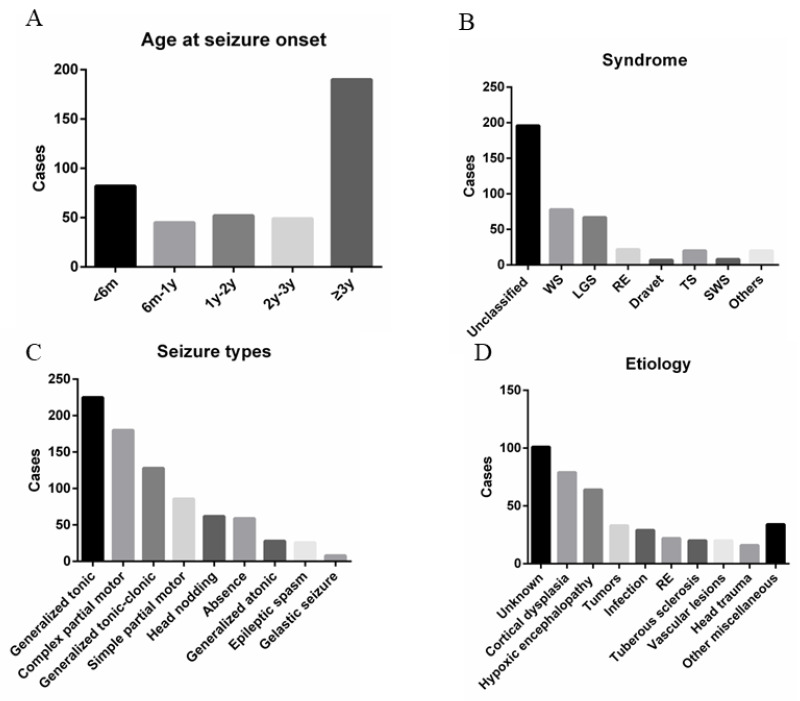
(**A**–**D**) summarized the clinical characteristics of pediatric patients with drug-resistant epilepsy in terms of age at seizure onset, epileptic syndrome, seizure type, and etiology, respectively.

**Figure 3 brainsci-13-00014-f003:**
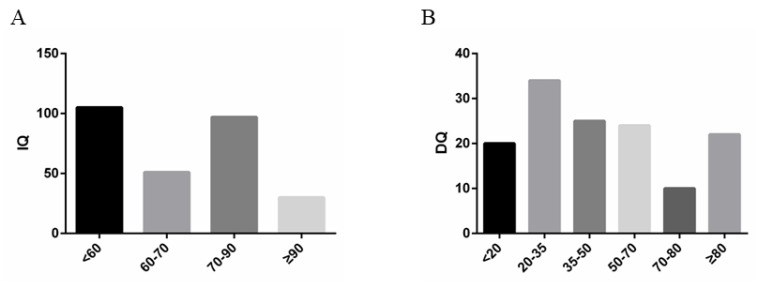
(**A**). IQ of children over 6 years old at enrollment. (**B**). DQ of children under 6 years of age at enrollment.

**Figure 4 brainsci-13-00014-f004:**
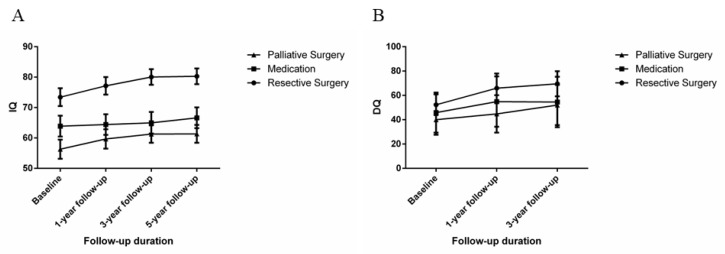
(**A**,**B**) The changes in IQ and DQ of children with drug-resistant epilepsy after different treatment regimens, respectively. It was worth noting that the Chinese version of the Gesell Developmental Schedules was only suitable for children under 6 years of age. Here we only listed the developmental quotients of 63 children who were suitable for observing the changes at follow-up.

**Figure 5 brainsci-13-00014-f005:**
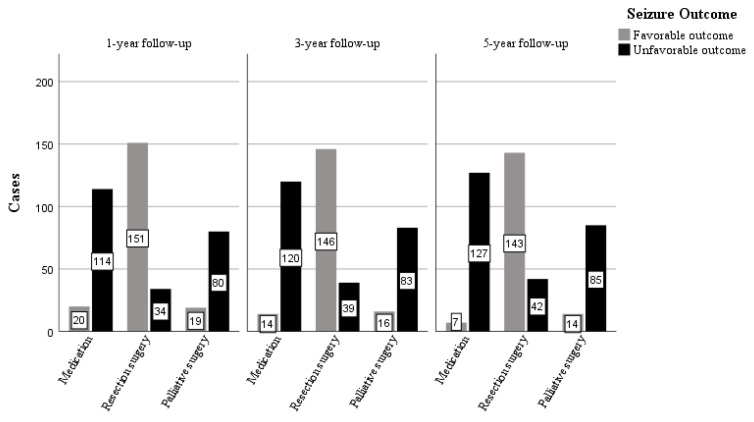
Number of patients with seizure outcomes according to different treatment regimens at different follow-up times.

**Table 1 brainsci-13-00014-t001:** The comparison of clinical characteristics between AEDs group and palliative surgery group (*n* = 233).

Clinical Characteristics	AEDs	Palliative Surgery	*p*
Gender, *n* (%)			0.629
Male	92 (39.5)	65 (27.9)	
Female	42 (18.0)	34 (14.6)	
Age	8.02 ± 3.69	8.20 ± 3.86	0.718
Age at seizure onset	3.17 ± 3.14	3.31 ± 3.12	0.723
Duration of seizures	4.86 ± 3.20	4.86 ± 2.77	0.992
Duration of AEDs	38.48 ± 37.55	41.71 ± 37.93	0.519
Seizure types, *n* (%)			0.275
Partial seizure only	8 (3.4)	7 (3.0)	
Generalized seizure only	59 (25.3)	53 (22.7)	
Generalized and partial seizure	67 (28.8)	39 (16.7)	
Brain MRI, *n* (%)			0.054
Normal or basal ganglia	33 (14.2)	15 (6.4)	
Unilateral	26 (11.2)	31 (13.3)	
Bilateral	75 (32.2)	53 (22.7)	
Interictal EEG, *n* (%)			0.095
Unilateral	24 (10.3)	10 (4.3)	
Bilateral	110 (47.2)	89 (38.2)	
Ictal onset rhythms, *n* (%)			0.492
Not captured	18 (7.7)	10 (4.3)	
Unilateral	12 (5.2)	6 (2.6)	
Bilateral	104 (44.6)	83 (35.6)	
PET, *n* (%)			0.715
No	117 (50.2)	88 (37.8)	
Yes	17 (7.3)	11 (4.7)	
MEG, *n* (%)			0.745
No	99 (42.5)	75 (32.2)	
Yes	35 (15.0)	24 (10.3)	
Invasive EEG, *n* (%)			0.166
No	133 (57.1)	95 (40.8)	
Yes	1 (0.4)	4 (1.7)	
Seizure outcome, *n* (%)			0.019*
Favorable outcome	7 (3.0)	14 (6.0)	
Unfavorable outcome	127 (54.5)	85 (36.5)	
Etiology (%)			0.560
Unknown	59 (25.3)	42 (18.0)	
Other miscellaneous	39 (16.7)	27 (11.6)	
Infection	9 (3.9)	12 (5.2)	
Hypoxic encephalopathy	27 (11.6)	18 (7.7)	

AED: antiepileptic drugs; MRI, magnetic resonance imaging; EEG: electroencephalogram; PET, positron emission tomography; MEG, Magnetoencephalography. (%), percentage of the total values. * *p* < 0.05.

**Table 2 brainsci-13-00014-t002:** Clinic characteristics of patients in resection surgery group and their relationship with seizure outcomes (*n* = 185).

Variable	Univariate Analysis		
	Favorable Outcomes	Unfavorable Outcomes	*p*
Gender, *n* (%)			
Male	92 (49.7)	23 (12.4)	0.261
Female	51 (27.6)	19 (10.3)	
Age at surgery	8.19 ± 3.87	8.93 ± 4.00	0.281
Age at seizure onset	4.42 ± 3.40	3.18 ± 2.80	0.032 *
Duration of seizures	3.78 ± 2.91	5.76 ± 3.60	0.009 *
Duration of AEDs	29.80 ± 25.97	43.00 ± 36.30	0.021 *
Seizure types, *n* (%)			0.200
Partial seizure only	49 (26.5)	10 (5.4)	
Generalized seizure only	30 (16.2)	14 (7.6)	
Generalized and partial seizure	64 (34.6)	18 (9.7)	
Brain MRI, *n* (%)			0.254
Normal or basal ganglia	5 (2.7)	4 (2.2)	
Unilateral	103 (55.7)	27 (14.6)	
Bilateral	35 (18.9)	11 (5.9)	
Interictal EEG, *n* (%)			0.943
Unilateral	57 (30.8)	17 (9.2)	
Bilateral	86 (46.5)	25(13.5)	
Ictal onset rhythms, *n* (%)			0.021 *
Not captured	20 (10.8)	3 (1.6)	
Unilateral	78 (42.2)	16 (8.6)	
Bilateral	45 (24.3)	23 (12.4)	
Invasive EEG, *n* (%)			0.088
No	127 (68.6)	33 (17.8)	
Yes	16 (8.6)	9 (4.9)	
PET, *n* (%)			0.857
No	107 (57.8)	32 (17.3)	
Yes	36 (19.5)	10 (5.4)	
MEG, *n* (%)			0.158
No	79 (42.7)	18 (9.7)	
Yes	64 (34.6)	24 (13.0)	
Etiology, *n* (%)			0.292
FCD	52 (31.9)	19 (10.3)	
Tumors	29 (15.7)	4 (2.2)	
Hypoxic-ischemic encephalopathy	13 (7.0)	6 (3.2)	
Others	49 (26.5)	13 (7.0)	

MRI, magnetic resonance imaging; EEG: electroencephalogram; PET, positron emission tomog-raphy; MEG, Magnetoencephalography; FCD: focal cortical dysplasia. (%), percentage of the total values. * *p* < 0.05.

**Table 3 brainsci-13-00014-t003:** Predictors of seizure outcomes in the resection surgery group for DRE on multivariate analysis (*n* = 185).

Variables	OR	95%CI	*p*
Age at seizure onset	0.948	0.840–1.069	0.382
Duration of seizures	1.299	1.115–1.513	0.001 *
Duration of AEDs	0.996	0.982–1.009	0.544
Ictal onset rhythms			
Not captured	Ref.		
Unilateral	4.099	1.075–15.636	0.039 *
Bilateral	2.543	0.695–9.300	0.158
Invasive EEG	0.882	0.389–2.002	0.764
MEG	1.494	0.742–3.006	0.261

OR: odds ratio; CI: confidence interval; AED: antiepileptic drugs; EEG: electroencephalogram; Ref. reference; MEG, magnetoencephalography. * *p* < 0.05.

## Data Availability

The data presented in this study are available on request from the corresponding author.

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
