# Peer review of "Retrospective Clinical Analysis of Epilepsy Treatment for Children with Drug-Resistant Epilepsy (A Single-Center Experience)"

_brainsci, 2022, doi:10.3390/brainsci13010014_

Round 1

Reviewer 1 Report

At the manuscript “Retrospective clinical analysis of epilepsy treatment for children with drug-resistant epilepsy (a single-center experience)” by Drs. Changqing Liu et al, authors describing results of retrospective cohort study of the clinical characteristics and seizure outcomes of patients aged 1–14 years with drug-resistant epilepsy, who were treated by different therapy’ protocols. There were three groups of patients: drug treatment, resection surgery and palliative surgery groups. The results obtained are based on a wide sample and are statistically significant. The most common types of epilepsy were identified. The authors concluded that pediatric patients with drug-resistant epilepsy usually characterized by frequent seizures, a variety of seizure types, and have complex etiology. Early surgical intervention is often helpful.

The manuscript fully meets the stated goals, contains extensive literature, and is written clearly. I have no objections to the essence of the manuscript.

I have only minor notes:

1. Lennox-Gastaut Syndrome, which the authors write about, has raised additional questions in recent years. I would suggest to add a couple of sentences about it and perhaps give a reference to one of the modern works, for example:

Nicholas J Evans , Krishna Das; Lennox Gastaut Syndrome - A strategic shift in diagnosis over time? Seizure;  2022 Dec;103:68-71. doi: 10.1016/j.seizure.2022.10.020.  

2. I would explain what is meant by the term stereotactic radiofrequency ablation - which structures were ablated, unilaterally or in both hemispheres?

Author Response

We thank you for the time and effort dedicated to the thorough review of the manuscript and for the detailed and specific comments that have helped considerably improve our research presentation. We have carefully revised our manuscript as suggested. The line numbers in the responses refer to the relevant parts of the main manuscript (changes marked) where revisions have been undertaken.

At the manuscript “Retrospective clinical analysis of epilepsy treatment for children with drug-resistant epilepsy (a single-center experience)” by Drs. Changqing Liu et al, authors describing results of retrospective cohort study of the clinical characteristics and seizure outcomes of patients aged 1–14 years with drug-resistant epilepsy, who were treated by different therapy’ protocols. There were three groups of patients: drug treatment, resection surgery and palliative surgery groups. The results obtained are based on a wide sample and are statistically significant. The most common types of epilepsy were identified. The authors concluded that pediatric patients with drug-resistant epilepsy usually characterized by frequent seizures, a variety of seizure types, and have complex etiology. Early surgical intervention is often helpful.

The manuscript fully meets the stated goals, contains extensive literature, and is written clearly. I have no objections to the essence of the manuscript.

I have only minor notes:

  1. Lennox-Gastaut Syndrome, which the authors write about, has raised additional questions in recent years. I would suggest to add a couple of sentences about it and perhaps give a reference to one of the modern works, for example:

Nicholas J Evans , Krishna Das; Lennox Gastaut Syndrome - A strategic shift in diagnosis over time? Seizure;  2022 Dec;103:68-71. doi: 10.1016/j.seizure.2022.10.020. 

Authors’ Response: We thank the reviewer for their positive comments on our work. Lennox-Gastaut syndrome is another common childhood epilepsy syndrome. Epilepsy syndromes in children are often refractory to AEDs. Intellectual or psychosocial dysfunction is inevitable in most children. We added a few lines of LGS as suggested by the reviewer. In addition, there is a point of view in the reviewer-recommended article that we agree with very much. This point of view is that the diagnosis of LGS should not be so strict. Once a patient's symptoms meet all diagnostic criteria, the patient's condition progresses. Thus, broader diagnostic criteria would allow patients to be treated earlier. Change to Text: Lines 270-280

  1. I would explain what is meant by the term stereotactic radiofrequency ablation - which structures were ablated, unilaterally or in both hemispheres?

Author Response: Thank you for your suggestion. First, we apologize for the expression stereotaxic radiofrequency ablation as it is not a common term and may be misunderstood. We replaced "stereotactic radiofrequency ablation" with "radiofrequency thermocoagulation" in the appropriate section. Furthermore, we briefly describe the reasons and candidates for palliative surgery with RF-TC. By retrospective analysis, patients with diffuse or multifocal brain abnormalities or negative imaging tended to be scheduled for RF-TC. The structure of RF-TC is based on the origin and propagation of epileptiform discharges.

Change to Text: Lines 127-132

Reviewer 2 Report

I suspect you may be working in a highly specialised, perhaps tertiary level, paediatric neurosurgery service and that this may have determined the spectrum of the seizure disorders in the children whose records were available. In your Methods section you have certainly provided information about this, but it does not appear elsewhere, even briefly, and if the paper is not read carefully readers may wonder why, for instance, drug resistant genetic generalised epilepsies are not mentioned anywhere. Also your inclusion criteria seemed to include the threat or actual existence of intellectual or psychological problems, and I wondered if this needed to be brought out more clearly as it may limit comparability of your results with other studies. Perhaps a few added words about the source of the patients studied or the inclusion criteria, in the Abstract of the paper would make the situation clear from the outset, and also make your findings more readily comparable with other published material.

I was not clear why an appreciable proportion of your patients did not go on to surgery. Were they simply considered not suited for it?

You write repeatedly of ‘favourable’ responses, but do not define what you mean by a favourable response. Are you referring to complete remission of seizures and no antiseizure medication use, or to something less than that?

There is very little information provided about antiseizure medication therapy. Did you have evidence that it had been tried adequately and appropriately prior to referral to your service? Did it continue to be managed optimally in children who did not proceed to surgery? Was it continued, perhaps modified, in those who underwent surgery so that a favourable response could be response to changed medical treatment in someone who had received surgery?

In your Tables I had difficulty in understanding some of the numbers in brackets until I realised that the percentage figures within brackets are not, as usually seems to be the practice, percentages of other values in the same column of the Table, but percentage of the total values in the same row of the Table. I wonder if you should explain this in the Legends to the Tables to avoid confusing readers.

Also, in the left-hand column of the Tables the centring of the text does not make it obvious where the items in two or three of the rows are really subsidiary to the item in the row above. To use some device, e.g. a different typeface, indenting with the centering function not used, would make the contents easier for readers to follow, and also easier to reconcile with the P values which can otherwise look as though they might be misplaced.

Author Response

We thank you for the time and effort dedicated to the thorough review of the manuscript and for the detailed and specific comments that have helped considerably improve our research presentation. We have carefully revised our manuscript as suggested. The line numbers in the responses refer to the relevant parts of the main manuscript (changes marked) where revisions have been undertaken.

I suspect you may be working in a highly specialised, perhaps tertiary level, paediatric neurosurgery service and that this may have determined the spectrum of the seizure disorders in the children whose records were available. In your Methods section you have certainly provided information about this, but it does not appear elsewhere, even briefly, and if the paper is not read carefully readers may wonder why, for instance, drug resistant genetic generalised epilepsies are not mentioned anywhere. Also your inclusion criteria seemed to include the threat or actual existence of intellectual or psychological problems, and I wondered if this needed to be brought out more clearly as it may limit comparability of your results with other studies. Perhaps a few added words about the source of the patients studied or the inclusion criteria, in the Abstract of the paper would make the situation clear from the outset, and also make your findings more readily comparable with other published material.

Authors’ Response: Thank you very much for your careful review and for pointing out the above-mentioned problems. Epilepsy is one of the most common disorders of neurological dysfunction. The causes of seizures are diverse and are summarized by ILAE as genetic, structural or metabolic and unknown. Our description of the relevant content in this paper was really insufficient, so we introduced additional information in the revised version. This study was a retrospective cohort study, which limited the completeness of some information, for example, whether seizures in these children were associated with epileptogenic genes. In addition, we highly appreciated your comments on the inclusion criteria. We overemphasized intellectual and psychological problems. The special feature of childhood epilepsy compared to adult epilepsy was that seizures might affect the development of the children’s brain. A consequence of this was deterioration of the patients’ cognitive function. Therefore, we have deleted the second inclusion criterion on cognitive impairment.

Change to Text: Introduction section; inclusion criteria

I was not clear why an appreciable proportion of your patients did not go on to surgery. Were they simply considered not suited for it?

Authors’ Response: Thank you for your comment. One of the main objective of this study was to compare the seizure outcome of different treatment in children with DRE. For patients with epileptogenic regions involving eloquent cortical areas or diffuse lesions or with negative imaging, seizure foci seemed to be unremovable. In this case, palliative surgery and adjustmen of the drug regimen were options. Previous studies showed that the probability of achieving seizure-free by tweaking their drug regimen after failure of two AEDs is less than 10%. Patients who continued to choose drug therapy were most likely to do so because of financial problems. It was for this reason that we had the opportunity to compare the seizure outcomes of palliative surgery and medication in children with DRE. In addition, 26 patients in the medication group were excluded because, although they were subsequently treated with surgery, they were not followed up long enough or did not have the procedure performed at our center. This study provided objective confirmation that palliative surgery was more effective than medication, which would be helpful in further promoting surgery for DRE.

You write repeatedly of ‘favourable’ responses, but do not define what you mean by a favourable response. Are you referring to complete remission of seizures and no antiseizure medication use, or to something less than that?

Authors’ Response: Thank you for you feedback. The definition of the “favorable seizure outcome” was presented in the “Materials and Methods” section, under the “Demographic and clinical characteristics” heading. In addition, we have listed the ILAE classification criteria for seizure outcomes in this section. The classification was based on seizure outcome and didn’t take into account the use of AEDs. In addition, although it was not appropriate to apply this classification criterion for pharmacotherapy, we also evaluated patients in the medication group by this criterion in order to make the results of the three groups of seizure outcomes comparable.

Change to Text: lines 118-124

There is very little information provided about antiseizure medication therapy. Did you have evidence that it had been tried adequately and appropriately prior to referral to your service? Did it continue to be managed optimally in children who did not proceed to surgery? Was it continued, perhaps modified, in those who underwent surgery so that a favourable response could be response to changed medical treatment in someone who had received surgery?

Authors’ Response: Thank you for your comment. As we mentioned in the inclusion criteria, all patients in this study were treated with AEDs, adrenocorticotropic hormone therapy or ketogenic diet prior to enrollment and their seizures were still not effectively controlled. The International League Against Epilepsy (ILAE) defined DRE as failure of adequate trials of two tolerated and appropriately chosen and used AED schedules (whether as monotherapies or in combination) to achieve sustained seizure freedom. The effectiveness of surgical treatment was only one aspect. Any patient who underwent surgery would also need to take AEDs. The decision to discontinue medication was based both on the patient's seizure outcome and on the patient's EEG review and seizure type. We have made a collection of the numbers of AEDs at the time of admission and at the last follow-up for each group of patients and have added the relevant content to this section of the results. In addition, 43 and 7 patients in resection group and palliative group achieved seizure free and weaned off AEDs. However, it was difficult to analyze whether there was a change in the dose of AEDs, as the required dose changed with age.

Change to Text: lines 198-203

In your Tables I had difficulty in understanding some of the numbers in brackets until I realised that the percentage figures within brackets are not, as usually seems to be the practice, percentages of other values in the same column of the Table, but percentage of the total values in the same row of the Table. I wonder if you should explain this in the Legends to the Tables to avoid confusing readers.

Authors’ Response: Thank you for pointing out this deficiency. We apologize for the inconvenience of your reading. According to your hints, we also realize that such a percentage expression can be confusing for the readers. We have modified the percentages in the brackets. The number in bracket represented the percentage of the number in front of the bracket to the total value.

Change to Text: Tables

Also, in the left-hand column of the Tables the centring of the text does not make it obvious where the items in two or three of the rows are really subsidiary to the item in the row above. To use some device, e.g. a different typeface, indenting with the centering function not used, would make the contents easier for readers to follow, and also easier to reconcile with the P values which can otherwise look as though they might be misplaced.

Authors’ Response: Thank you for your suggestion. According to the Reviewer’s suggestion, we  have eliminated the centering performance of the content in the left column of the table. Not only that, the placement of the tables and figures is rather messy, we have transfered them to the corresponding results section for the convenience of the reader. This is the first time we have submitted the article in this journal system. It proved that our typesetting skills should be improved.

Change to Text: Tables

Reviewer 3 Report

This study represented retrospective clinical analysis of epilepsy treatment for children with drug-resistant epilepsy from a single center. There are some issues in this manuscript that should be addressed as follows:

·         Abstract:

- The subheadings of the abstract shouldn’t be numbered.

- Page 1 Line 14: The word “Objective” should be replaced with “Objectives”.

·         Introduction:

1.    The introduction is too short for a common subject like epilepsy. The predisposing factors of DRE should be clarified and lines of management of this condition should be mentioned.

2.    The novel aspects of this study should be mentioned as there are similar studies concerning the same topic.

·         Results:

  1. Tables 2 and 3 and figures 4 and 5 should be transferred and mentioned in the text of the “Results” section.
  2. A collective diagram summarizing the main findings of this study is recommended.

·         Discussion: The discussion should be summarized to focus on analysis of the results of the present study.

·         Conclusion: The clinical implications of the results of the present study should be mentioned.

·         General comments:  

1. The manuscript should be revised by English-naïve speaker to improve the quality of the language.

2. The manuscript should be checked regarding the grammatical errors and plagiarism.

3. The code of approval of the research ethics committee should be provided.

Author Response

We thank you for the time and effort dedicated to the thorough review of the manuscript and for the detailed and specific comments that have helped considerably improve our research presentation. We have carefully revised our manuscript as suggested. The line numbers in the responses refer to the relevant parts of the main manuscript (changes marked) where revisions have been undertaken.

This study represented retrospective clinical analysis of epilepsy treatment for children with drug-resistant epilepsy from a single center. There are some issues in this manuscript that should be addressed as follows:

  • Abstract:

- The subheadings of the abstract shouldn’t be numbered.

Authors’ Response: We deeply appreciate the Reviewer’s suggestion. According to the Reviewer’s comment, we have deleted the numbers in the abstract.

Change to Text: lines 14

- Page 1 Line 14: The word “Objective” should be replaced with “Objectives”.

Authors’ Response: We are extremely grateful to reviewer for pointing out this problem. We have replaced “Objective” with “Objectives” in the corresponding part.

Change to Text: Abstract

  • Introduction:
  1. The introduction is too short for a common subject like epilepsy. The predisposing factors of DRE should be clarified and lines of management of this condition should be mentioned.

Authors’ Response: Thank you for underlining this deficiency. This section was revised and modified according to the information showed in the work suggested by the Reviewer. We have added in this section the incidence, common causes, and current status of clinical management of DRE in children.

Change to Text: lines 39-42; 48-59;

  1. The novel aspects of this study should be mentioned as there are similar studies concerning the same topic.

Authors’ Response: Thank you for your suggestion. As suggested by Reviewer, we have added the suggested content to the manuscript on page 2.

Change to Text: lines 66-73

  • Results:

Tables 2 and 3 and figures 4 and 5 should be transferred and mentioned in the text of the “Results” section.

Authors’ Response: Thank you for your comments. We are sorry for the inconvenience they caused in your reading. This is our first submission in this system. As a result is that we really have no relevant experience in layout. We have transferred the tables and figures to the appropriate positions as requested by Reviewer.

Change to Text: Tables 2 and 3 and figures 4 and 5

A collective diagram summarizing the main findings of this study is recommended.

Authors’ Response: Thank you for your suggestion. A collective diagram could provide an overview of the main findings of this study. However, some of the authors of this article believe that the existing figures and tables can summarize the results. Although the diagram is more concise and clear, they may also cause the readers to miss some results of interest to them. Thank you again for your suggestion.

  • Discussion: The discussion should be summarized to focus on analysis of the results of the present study.

Authors’ Response: Thank you for your comment. As suggested by Reviewer, we have modified the discussion section. The present study was a clinical retrospective cohort research. Basic medical research was discussed and cited excessively. Therefore, we have deleted the relevant content. In addition, we also have added statistics on the number of AEDs applied to patients in the results section, which is analyzed and discussed.

Change to Text: Discussion section.

  • Conclusion: The clinical implications of the results of the present study should be mentioned.

Authors’ Response: Thank you for your suggestion. According to the Reviewer’s comment, we have modified the conclusion section. This study summarized the clinical features of drug-refractory epilepsy in children. The seizure outcomes of different treatment were compared, aiming to further guide clinical practice and facilitate timely and effective treatment of pediatric patients.

Change to Text: lines 29-33; 350-353

  • General comments:  
  1. The manuscript should be revised by English-naïve speaker to improve the quality of the language.

Authors’ Response: Thank you for your careful review. We are very sorry for the inconvenience in your reading. The manuscript has been thoroughly revised and rewritten by a native English speaker, so we hope it can meet the journal’s standard.

  1. The manuscript should be checked regarding the grammatical errors and plagiarism.

Authors’ Response: Thank you for your suggestion. This manuscript has been revised by a native English speaker. We hereby declare that there is no plagiarism in this paper. For parts of the content that are similar, we have marked the references.

  1. The code of approval of the research ethics committee should be provided.

Authors’ Response: Thank you for your careful review. This study was approved by the ethics committee, which was demonstrated in the materials and methods section under the “Patient selection” heading. Due to our oversight no code of approval of the research ethics committee was marked. We have made corrections.

Change to Text: line 372